

# Genetic diversity and population structure of the endangered basal angiosperm *Brasenia schreberi* (Cabombaceae) in China

Zhi-Zhong Li[1,2], Andrew W. Gichira[1,2,3], Qing-Feng Wang[2,3] and Jin-Ming Chen[2]

[1] University of Chinese Academy of Sciences, Beijing, China
[2] Key Laboratory of Aquatic Botany and Watershed Ecology, Wuhan Botanical Garden, Chinese Academy of Sciences, Wuhan, China
[3] Sino-Africa Joint Research Center, Chinese Academy of Sciences, Wuhan, China

## ABSTRACT

*Brasenia schreberi* J.F. Gmelin (Cabombaceae), an aquatic herb that occurs in fragmented locations in China, is rare and endangered. Understanding its genetic diversity and structure is crucial for its conservation and management. In this study, 12 microsatellite markers were used to estimate the genetic diversity and variation in 21 populations of *B. schreberi* in China. A total of 61 alleles were found; assessment of allelic richness ($Ar = 1.92$) and observed and expected heterozygosity ($H_O = 0.200$, $H_E = 0.256$) suggest lower genetic diversity compared to some endangered species, and higher variation was observed within populations (58.68%) rather than among populations (41.32%). No significant correlation between geographical and genetic distance among populations was detected (Mantel test, $r = 0.0694$; $P = 0.7985$), which may have likely resulted from barriers to gene flow ($Nm = 0.361$) that were produced by habitat fragmentation. However, Bayesian and neighbor-joining cluster analyses suggest a population genetic structure consisting of two clusters (I and II) or four subclusters (I-1, 2 and II-1, 2). The genetic structure and distribution of *B. schreberi* in China may have involved glacial refugia that underwent range expansions, introgression, and habitat fragmentation. The findings of the present study emphasize the importance for both in situ and ex situ conservation efforts.

## INTRODUCTION

*Brasenia schreberi* J.F. Gmelin, a basal angiosperm that belongs to family Cabombaceae (Nymphaeales), is a perennial aquatic plant. Similar to most aquatic plants, *B. schreberi* can reproduce sexually by outcrossing or asexually through rhizomes and winter buds (*Bertin, 1993*; *Griffin, Mavraganis & Eckert, 2000*). It has a wide yet and sporadic geographical distribution in temperate and tropical regions of Asia, Australia, Africa, India, and North and South America (*Kim et al., 2012*).

Prior to the early 20th century, *B. schreberi* was widely distributed in China and grew in unpolluted aquatic environments such as freshwater ponds, lakes, swamps, and even

Corresponding authors
Zhi-Zhong Li,
wbg_georgelee@163.com
Jin-Ming Chen, jmchen@wbgcas.cn

agricultural fields. However, in recent decades, its population has significantly decreased due to the loss of natural habitats and deterioration in water quality resulting from excessive human activities, particularly involving leaves harvested for food and increasing the use of fertilizers and pesticides. Previous field investigations experienced difficulty in finding natural populations in regions within China where *B. schreberi* was previously known to grow in abundance (*Gao, Zhang & Chen, 2007*). Similar situations have been reported in other countries such as Korea and Japan (*Kim et al., 2012*). It is currently listed as a critically endangered species in China, belonging to the first category of key protected wild plants (*Yu, 1999*). Therefore, effective conservation management to preserve the remaining populations of *B. schreberi* is imperative.

Demonstration of genetic diversity and structure in rare plant species is often crucial for formulation of conservation and management strategies because it provides valuable insights into the vital aspects of demography, reproduction, and ecology (*Zaya et al., 2017*). Previous studies have employed various molecular markers to assess the genetic diversity of *B. schreberi*, including inter-simple sequence repeat markers (*Zhang & Gao, 2008*), randomly amplified polymorphic DNA, amplified fragments length polymorphisms (*Kim, Na & Choi, 2008*), and nuclear ribosomal spacer and chloroplast DNA sequences (*Kim et al., 2012*). Despite the significant decrease in the size of *B. schreberi* populations, these studies reveal significant genetic diversity, highlight factors that negatively influence genetic diversity, and propose potential conservation measures. *Zhang & Gao (2008)* investigated the population diversity in semi-natural populations of *B. schreberi* in Zhejiang and Suzhou provinces, whereas *Kim et al. (2012)* focused on natural populations from South Korea. However, no research investigations on the level and pattern of diversity and genetic structure in the wild populations of *B. schreberi* in China have been conducted to date.

Compared to other molecular markers, simple sequence repeat (SSR) markers have numerous merits, including co-dominance, high reproducibility, a relatively high level of polymorphisms, and are plentiful in the genome (*Liao et al., 2013*). SSR markers have been successfully applied to estimate the genetic diversity of other aquatic plants such as *Sparganium emersum* (*Pollux et al., 2007*), *Zostera marina* (*Reusch, Stam & Olsen, 2000*; *Talbot et al., 2016*), *Sagittaria natans* (*Yue et al., 2011*), *Zizania latifolia* (*Chen et al., 2012, 2017*), *Nymphoides peltata* (*Liao et al., 2013*), *Isoetes hypsophila* (*Li et al., 2013*), *Ruppia cirrhosa* (*Martínez-Garrido, González-Wangüemert & Serrão, 2014*), *Nuphar submersa* (*Shiga et al., 2017*), and *Ottelia acuminata* (*Zhai, Yin & Yang, 2018*). Here, we used 12 microsatellite loci to detect and estimate the genetic variation of *B. schreberi* in China. In this study, a total of 21 populations, representing nearly the entire known natural distribution zones in China, were sampled. We aimed to determine (1) how the extent of genetic diversity is apportioned within and among populations of *B. schreberi*, and (2) the genetic structure and its association with geographical distribution. The findings of the present study may be utilized in conservation efforts on this species.
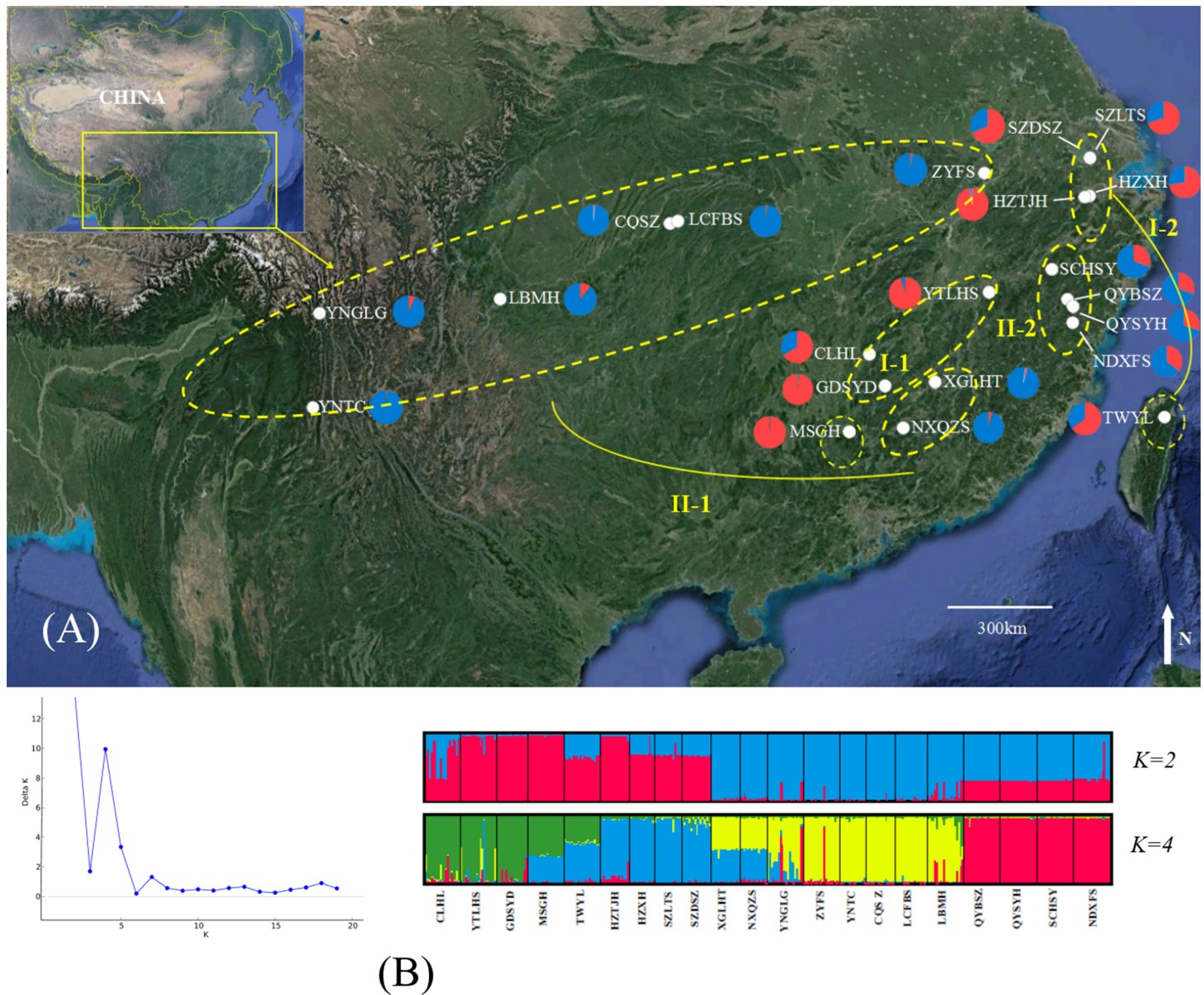

**Figure 1  Genetic structure of *B. schreberi* populations in China.** (A) Sampling area and genetic structure of *B. schreberi* populations based on *K* = 2 genetic clusters (I and II); (B) proportional membership of 21 *B. schreberi* populations to *K* = 2 and 4 (subcluster I-1, I-2, II-1, II-2) genetic clusters. Individuals are represented by a single vertical column divided into two or four (=*K*) colors. The relative length of the colored segment corresponds to the individual's estimated proportion of membership in that cluster. Map data © 2016 Google.

## MATERIALS AND METHODS

### Sample collection

During June–September 2016, fresh leaves of 376 individuals from 21 locations across almost the entire geographical distribution of *B. schreberi* in China were collected (Fig. 1) and rapidly dried in silica gel until further analyses. We sampled a range of 14–20 individuals per site (Table 1). Because *B. schreberi* could reproduce asexually through

**Table 1** Location of populations and number of samples in each population of *B. schreberi* in China.

| Population | Location | Latitude/Longitude | Sample size | Voucher code |
|---|---|---|---|---|
| HZXH | Hangzhou, Zhejiang | 30°08′N/120°13′E | 14 | HIB-BS08 |
| XGLHT | Ganzhou, Jiangxi | 26°04′N/115°21′E | 16 | HIB-BS06 |
| CQSZ | Shizhu, Chongqing | 30°09′N/108°29′E | 16 | HIB-BS01 |
| HZTJH | Hangzhou, Zhejiang | 30°07′N/120°04′E | 16 | HIB-BS02 |
| NXQZS | Nanxiong, Guangdong | 25°03′N/114°25′E | 15 | HIB-BS11 |
| SZDSZ | Suzhou, Jiangsu | 31°04′N/120°24′E | 16 | HIB-BS15 |
| SZLTS | Suzhou, Jiangsu | 31°03′N/120°24′E | 15 | HIB-BS10 |
| YNGLG | Gaoligong, Yunnan | 28°01′N/98°38′E | 20 | HIB-BS18 |
| YNTC | Tengchong, Yunnan | 25°44′N/98°34′E | 14 | HIB-BS19 |
| MSGH | Mangshan, Hunan | 25°03′N/113°00′E | 20 | HIB-BS12 |
| YTLHS | Yingtan, Jiangxi | 28°04′N/117°02′E | 20 | HIB-BS22 |
| LBMH | Leibo, Sichuan | 28°26′N/103°48′E | 20 | HIB-BS26 |
| CLHL | Chaling, Hunan | 26°50′N/113°40′E | 19 | HIB-BS29 |
| GDSYD | Guidong, Zhejiang | 26°04′N/114°01′E | 17 | HIB-BS36 |
| QYBSZ | Qingyuan, Zhejiang | 27°43′N/119°11′E | 20 | HIB-BS23 |
| QYSYH | Qingyuan, Zhejiang | 27°32′N/119°19′E | 20 | HIB-BS07 |
| SCHSY | Suichang, Zhejiang | 28°28′N/118°51′E | 20 | HIB-BS09 |
| NDXFS | Ningde, Fujian | 27°09′N/119°15′E | 20 | HIB-BS14 |
| ZYFS | Zongyang, Anhui | 30°55′N/117°17′E | 20 | HIB-BS17 |
| LCFBS | Lichuan, Hubei | 30°12′N/108°42′E | 18 | HIB-BS20 |
| TWYL | Yiliang, Taiwan | 24°38′N/121°31′E | 20 | HIB-BSTW |
| Total | | | 376 | |

its stolons, we selected leaves from plants within populations that were separated by at least five metres to reduce the collection of clonal individuals. Voucher specimens from each population were deposited in the herbarium of Wuhan Botanical Garden, Chinese Academy of Sciences.

## DNA extraction, amplification, and sequencing

Total genomic DNA was extracted from silica-dried leaves using the modified cetyltrimethylammonium bromide method described by *Doyle & Doyle (1987)*. SSRs were genotyped using 12 polymorphic SSR loci from previous study (*Liu et al., 2016*) using polymerase chain reaction (PCR). The total reaction volume was 25 $\mu$L, which consisted of 0.25 mM of each dNTP, five $\mu$L of 10 × Taq buffer (10 mM Tris–HCl, (pH 8.3), 1.5 mM MgCl$_2$, and 50 mM KCl), one mM of each forward primer labeled with a fluorescent chemical and an unlabeled reverse primer, one U of Taq polymerase (TransGen Biotech Co., Beijing, China), and 10–30 ng of genomic DNA as template. PCR amplification was performed with a PTC-100 thermocycler (Bio-Rad, Hercules, CA, USA) using the following program profile: 95 °C for 3 min; followed by 30 cycles of 95 °C for 30 s, annealing at 55–60 °C for 30 s (depending on primer), and 72 °C for 40 s; and a final extension at 72 °C for 10 min. The PCR products were separated via electrophoresis on

1.0% (w/v) agarose gels, stained with ethidium bromide, and observed under UV light. Then, the multiplex amplified PCR products were sequenced on an ABI prism 3730xl and sized using an internal DNA standard (Rox-500; Life Technologies, Shanghai, China). The SSR fragments were visualized with GENEMAPPER v.4.0 (Applied Biosystems, Foster City, CA, USA).

## Data analysis

The loci were tested for Hardy–Weinberg equilibrium (HWE) and linkage disequilibrium using GENEPOP ver. 4.0 (*Rousset, 2008*). Deviations from HWE due to null alleles, stuttering, and large allele dropout were assessed using MICROCHECKER ver. 2.2.3 (*Van Oosterhout et al., 2004*).

The total number of alleles ($N$a), the effective number of alleles ($N$e), observed ($H$o) and expected ($H$e) heterozygosity, Shannon's information index ($I$), F-statistics ($F_{IS}$, $F_{ST}$), and gene flow ($N$m) were computed for each locus using POPGENE v.1.31 (*Yeh, Yang & Boyle, 1999*). The Polymorphic Information Content (PIC) for each locus was estimated using CERVUS ver. 3.0 (*Kalinowski, Taper & Marshall, 2007*).

For each population, the average number of alleles ($N_A$), $N_E$, the number of private allele ($A_P$), $H_O$, $H_E$, $I$ and, the pairwise $F_{ST}$ between each pair of populations were estimated using GENALEX ver. 6.0 (*Peakall & Smouse, 2006*). Allelic richness ($A$r) and inbreeding coefficients ($F_{IS}$) were calculated using the diveRsity (*Keenan et al., 2013*) packages and confidence intervals were estimated with 10,000 bootstrap replicates. Gene flow ($N$m) among populations was calculated using the expression $Nm = (1 - F_{ST})/4F_{ST}$ (*Slatkin, 1993*). Analysis of molecular variation was used to evaluate the relative level of genetic variations among groups ($F_{CT}$), among populations within groups ($F_{SC}$), and among individuals within populations ($F_{ST}$); these values, and the significance of each value, were tested using Arlequin ver. 3.5 (*Excoffier, Smouse & Quattro, 1992*; *Excoffier, Laval & Schneider, 2005*).

The neighbor-joining (NJ) tree was constructed using the software TreeFit (*Kalinowski, 2009*) based on Nei's genetic distance ($D_A$; *Nei, 1987*) to reveal the genetic relationships among the populations used in this study. The correlation between genetic distance and geographic distance was estimated using a Mantel test based on a matrix of genetic distance using $D_A$ and a pairwise matrix of geographic distance. Isolation by Distance Web Service ver. 3.23 (*Jensen, Bohonak & Kelley, 2005*) was used to test for significance with 10,000 permutations.

The BOTTLENECK program (*Piry, Luikart & Cornuet, 1999*) was used to estimate the possible influence of recent demographic changes on genetic diversity and identify genetic bottlenecks among populations. Wilcoxon signed-rank tests were conducted using three different models with 10,000 replicates: the infinite allele mutation model (IAM), the stepwise mutation model (SMM), the two-phased model of mutation (TPM) which with 70% of mutations were assumed to occur under the SMM, and 30% were assumed to occur under the IAM. The mode shift of each population was also estimated using BOTTLENECK using default settings.

**Table 2 Genetic diversity at 12 SSR loci in 376 individuals of *B. schreberi*.**

| Locus | Annealing temp (°C) | Allele size (bp) | Na | Ne | I | PIC | Ho | He | $F_{IS}$ | $F_{ST}$ | Nm |
|-------|------|------|------|------|------|------|------|------|------|------|------|
| BS02 | 59 | 147–159 | 3 | 1.268 | 0.438 | 0.483 | 0.019 | 0.212 | 0.777 | 0.516 | 0.234 |
| BS03 | 60 | 115–130 | 8 | 2.404 | 1.253 | 0.547 | 0.341 | 0.585 | −0.068 | 0.4444 | 0.313 |
| BS04 | 59 | 128–212 | 3 | 1.246 | 0.390 | 0.496 | 0.186 | 0.198 | −0.123 | 0.207 | 0.96 |
| BS05 | 60 | 273–291 | 13 | 5.710 | 1.952 | 0.854 | 0.388 | 0.826 | −0.254 | 0.61 | 0.16 |
| BS06 | 60 | 209–218 | 4 | 1.075 | 0.179 | 0.458 | 0.051 | 0.070 | −0.244 | 0.419 | 0.347 |
| BS08 | 60 | 138–160 | 7 | 3.051 | 1.277 | 0.740 | 0.154 | 0.673 | 0.590 | 0.486 | 0.264 |
| BS09 | 60 | 265–297 | 3 | 1.827 | 0.653 | 0.539 | 0.003 | 0.453 | 0.973 | 0.71 | 0.102 |
| BS15 | 60 | 231–268 | 4 | 1.223 | 0.400 | 0.465 | 0.016 | 0.183 | 0.905 | 0.181 | 1.129 |
| BS18 | 60 | 168–177 | 2 | 1.701 | 0.680 | 0.515 | 0.194 | 0.413 | 0.458 | 0.152 | 1.393 |
| BS19 | 60 | 263–289 | 7 | 3.116 | 1.283 | 0.747 | 0.955 | 0.680 | −0.706 | 0.176 | 1.173 |
| BS22 | 60 | 218–231 | 4 | 2.521 | 1.124 | 0.619 | 0.085 | 0.604 | 0.791 | 0.318 | 0.536 |
| BS29 | 60 | 230–270 | 3 | 1.574 | 0.559 | 0.623 | 0.003 | 0.365 | 0.987 | 0.492 | 0.258 |
| Mean | | | 5.167 | 2.226 | 0.849 | 0.591 | 0.200 | 0.438 | 0.341 | 0.393 | 0.572 |
| SD | | | 3.07 | 1.307 | 0.523 | 0.129 | 0.271 | 0.239 | 0.169 | 0.053 | 0.132 |

**Note:**
Na, the total number of alleles; Ne, the effective number of alleles; Ho, the observed heterozygosity; He, the expected heterozygosity; I, Shannon's information index; $F_{IS}$, coefficient of inbreeding; $F_{ST}$, genetic differentiation coefficient; PIC, polymorphic information content; Nm, gene flow.

The population structure was tested using STRUCTURE ver. 2.3.1 (*Pritchard, Stephens & Donnelly, 2000*) based on a Bayesian clustering method. The approach was used to cluster genetically similar individuals and assess the most likely clustering state under the Hardy–Weinberg principle. The optimum number of genetic clusters ($K$) was tested from $K = 2$ to 20 clusters based on assuming admixture, correlated allele frequencies. The analysis was performed for 10 iterations, with a burn-in of 80,000 replications, followed by 800,000 Markov Chain Monte Carlo replications. The most likely number of clusters was verified based on the $\Delta K$ method (*Evanno, Regnaut & Goudet, 2005*). To generate a consensus $K$, independent runs of all data were normalized with CLUMPP ver. 1.1.2 (*Jakobsson & Rosenberg, 2007*) using the Greedy algorithm with 100,000 repeats. DISTRUCT ver. 1.1 (*Rosenberg, 2004*) was used to visualize the population structure.

## RESULTS

### Microsatellite variation and genetic diversity within populations

Null alleles were observed in a few loci (BS02, BS06, BS09, BS18, and BS19). A total of 61 alleles were detected across 12 SSR loci in 376 individuals from 21 populations (Table 2). The number of alleles generated by each marker ranged from two at locus BS18 to 13 at locus BS05, with an average of 5,167 alleles at each locus. Ne for each locus ranged from 1.075 to 5.710. Ho ranged from 0.003 to 0.955, and He ranged from 0.07 to 0.826. High $F_{ST}$ values ($F_{ST}$ = 0.152–0.710, mean = 0.393) and PIC values (PIC = 0.483–0.854, mean = 0.591) were detected at all loci. The average $F_{IS}$ across all loci was 0.341. Nm per locus varied between 0.102 in locus BS09 and 1.393 in locus BS18 (Table 2).

**Table 3 Summary statistics for *B. schreberi* populations in China.**

| Population | $N_A$ | $N_E$ | Ar | Ap | $H_O$ | $H_E$ | $F_{IS}$ | I | IAM | TPM | SMM | Mode-shift |
|---|---|---|---|---|---|---|---|---|---|---|---|---|
| HZXH | 2.417 (0.434) | 1.796 (0.336) | 2.42 (0.78) | 0 | 0.310 (0.120) | 0.285 (0.085) | −0.049 (0.096) | 0.520 (0.161) | 0.313 | 0.742 | 0.844 | Shifted |
| XGLHT | 1.750 (0.218) | 1.374 (0.141) | 1.75 (0.35) | 0 | 0.104 (0.083) | 0.201 (0.065) | 0.507 (0.097) | 0.323 (0.100) | 0.469 | 0.469 | 0.938 | L-shaped |
| CQSZ | 2.250 (0.279) | 1.616 (0.179) | 2.23 (0.50) | 0 | 0.177 (0.100) | 0.303 (0.068) | 0.442 (0.230) | 0.500 (0.111) | 0.250 | 0.359 | 0.910 | L-shaped |
| HZTJH | 2.000 (0.369) | 1.420 (0.176) | 1.93 (0.66) | 0 | 0.266 (0.128) | 0.193 (0.076) | −0.349 (0.176) | 0.318 (0.124) | 1.000 | 0.844 | 0.438 | L-shaped |
| NXQZS | 1.750 (0.218) | 1.484 (0.157) | 1.74 (0.44) | 0 | 0.256 (0.130) | 0.248 (0.069) | 0.005 (0.191) | 0.380 (0.106) | 0.039* | 0.054 | 0.375 | Shifted |
| SZDSZ | 2.333 (0.466) | 1.943 (0.343) | 2.32 (0.75) | 1 | 0.302 (0.125) | 0.318 (0.093) | 0.082 (0.114) | 0.552 (0.171) | 0.016* | 0.023* | 0.055 | Shifted |
| SZLTS | 2.333 (0.432) | 1.640 (0.246) | 2.32 (0.71) | 1 | 0.172 (0.093) | 0.266 (0.080) | 0.383 (0.156) | 0.468 (0.145) | 0.742 | 1.000 | 0.461 | L-shaped |
| YNGLG | 1.833 (0.271) | 1.412 (0.149) | 1.81 (0.48) | 0 | 0.113 (0.083) | 0.222 (0.064) | 0.512 (0.112) | 0.362 (0.107) | 0.039* | 0.039* | 0.578 | Shifted |
| YNTC | 1.917 (0.149) | 1.603 (0.143) | 1.92 (0.32) | 1 | 0.113 (0.078) | 0.319 (0.061) | 0.666 (0.173) | 0.481 (0.090) | 0.014* | 0.024* | 0.193 | Shifted |
| MSGH | 1.917 (0.193) | 1.482 (0.142) | 1.85 (0.44) | 1 | 0.213 (0.102) | 0.256 (0.066) | 0.196 (0.126) | 0.396 (0.096) | 0.203 | 0.496 | 0.652 | L-shaped |
| YTLHS | 2.667 (0.284) | 1.818 (0.189) | 2.58 (0.50) | 6 | 0.271 (0.084) | 0.398 (0.051) | 0.343 (0.201) | 0.660 (0.091) | 0.102 | 0.320 | 0.831 | Shifted |
| LBMH | 2.500 (0.337) | 1.860 (0.191) | 2.41 (0.51) | 3 | 0.233 (0.090) | 0.398 (0.062) | 0.435 (0.126) | 0.644 (0.110) | 0.005** | 0.024* | 0.275 | Shifted |
| CLHL | 1.917 (0.193) | 1.634 (0.146) | 1.92 (0.51) | 1 | 0.175 (0.073) | 0.328 (0.064) | 0.486 (0.114) | 0.500 (0.098) | 0.002** | 0.002** | 0.002** | Shifted |
| GDSYD | 1.667 (0.142) | 1.497 (0.141) | 1.65 (0.47) | 2 | 0.181 (0.093) | 0.259 (0.070) | 0.328 (0.130) | 0.372 (0.096) | 0.020* | 0.020* | 0.055 | Shifted |
| QYBSZ | 1.583 (0.193) | 1.237 (0.108) | 1.54 (0.45) | 0 | 0.163 (0.097) | 0.139 (0.056) | −0.145 (0.133) | 0.220 (0.081) | 1.000 | 1.000 | 1.000 | L-shaped |
| QYSYH | 1.500 (0.151) | 1.283 (0.114) | 1.49 (0.37) | 0 | 0.163 (0.101) | 0.162 (0.061) | 0.022 (0.187) | 0.243 (0.086) | 0.078 | 0.563 | 0.563 | L-shaped |
| SCHSY | 1.750 (0.218) | 1.390 (0.134) | 1.67 (0.45) | 0 | 0.163 (0.088) | 0.209 (0.068) | 0.245 (0.210) | 0.315 (0.099) | 0.375 | 0.375 | 0.578 | L-shaped |
| NDXFS | 1.500 (0.195) | 1.422 (0.157) | 1.50 (0.49) | 0 | 0.279 (0.118) | 0.207 (0.075) | −0.327 (0.190) | 0.301 (0.110) | 0.031* | 0.031* | 0.031* | Shifted |
| ZYFS | 1.917 (0.229) | 1.344 (0.143) | 1.84 (0.32) | 0 | 0.100 (0.082) | 0.186 (0.062) | 0.482 (0.145) | 0.312 (0.096) | 0.844 | 0.742 | 0.195 | L-shaped |
| LCFBS | 1.667 (0.188) | 1.321 (0.118) | 1.63 (0.40) | 0 | 0.093 (0.083) | 0.183 (0.062) | 0.515 (0.220) | 0.281 (0.089) | 0.469 | 0.578 | 0.937 | L-shaped |
| TWYL | 1.833 (0.167) | 1.547 (0.125) | 1.81 (0.42) | 1 | 0.354 (0.137) | 0.301 (0.061) | −0.153 (0.190) | 0.444 (0.086) | 0.006** | 0.006** | 0.020* | Shifted |
| Mean (SD) | 1.952 (0.060) | 1.530 (0.040) | 1.92 (0.03) | 0.81 (1.44) | 0.200 (0.022) | 0.256 (0.015) | 0.220 (0.031) | 0.409 (0.024) | | | | |

**Notes:**
Number in parentheses are standard deviations.
$N_A$, the average number of alleles; $N_E$, the effective number of alleles; Ar, the allelic richness; Ap, the private allele; $H_O$, the observed heterozygosity; $H_E$, the expected heterozygosity; PPL, percentage of polymorphic loci; $F_{IS}$, the inbreeding coefficient; I, Shannon's information index.
*/** Significant difference ($P < 0.1/0.05$).

The average number of alleles per population ranged from 1.5 ± 0.151 (QYSYH) to 2.667 ± 0.284 (YTLHS). Six private alleles, distinctive to a specific population, were observed in population YTLHS, three were recorded in population LBMH, and two were recorded in population GDSYD, whereas other populations, including SZDSZ, SZLTS, YNTC, MSGH, CLHL, and TWYL, had a single private allele each (Table 3). $Ar$ ranged from 1.49 to 2.58. $H_O$ and $H_E$ ranged from 0.093 ± 0.083 to 0.354 ± 0.137 and from 0.139 ± 0.056 to 0.398 ± 0.051, respectively. $I$ ranged from 0.220 ± 0.081 to 0.660 ± 0.091. $F_{IS}$ ranged from −0.349 to 0.666.

The Wilcoxon signed-rank tests revealed significant bottlenecks in 11 populations based on the IAM, SMM, and TPM assumptions, and normal models of distribution (L-shaped) for allelic frequencies were exhibited in other populations (Table 3).

## Genetic diversity and differentiation among populations

The $\Delta K$ values were calculated to assess the optimal number of genetic clusters ($K$) in the population structure. Analyses using STRUCTURE gave the highest value for two distinct genetic clusters ($K = 2$) and a second possibility was $K = 4$; both of these displayed biological significance (Fig. 1). We analyzed the bar-plot outputs using the best $K$-value, where $K = 4$ further indicated deep genetic structure. At $K = 2$, cluster $I$ included nine populations from Hunan Province (CLHL, GDSYD, MSGH), Zhejiang Province (HZXH, HZTJH), Jiangsu Province (SZDSZ, SZLTS), Jiangxi Province (YTLHS), and Taiwan (TWYL). At $K = 4$, each cluster defined above were further split into two subclusters. The rest of the populations were from cluster II, which were also separated into two subclusters (I-1, 2 and II-1, 2). Further population structure analysis indicated differentiation between two clusters (I and II) or among four subclusters (I-1, 2 and II-1, 2).

Under the $D_A$ distances (*Nei, 1987*), the NJ dendrogram (Fig. 2) indicated the genetic relationships among the studied populations (Table S2). Two main groups were also depicted, corroborating the STRUCTURE results, and each group had two subgroups. For example, the subgroup that included four populations (QYSYH, QYBSZ, SCHSY, and NDXFS) coincided with the results obtained from STRUCTURE; moreover, these populations were located in the same geographic region. These findings were suggestive of genetic differentiation among populations from southwest China, central China, and southeast China.

Analysis of molecular variation indicated substantial genetic variations among groups, populations, and individuals (Table 4). Of the total genetic diversity, the main components of genetic variation were from individuals within populations (58.68%), whereas 41.32% was attributable to variations among populations. When $K = 2/4$, differentiation was attributed to individuals within populations and among groups at 55.60%/56.19% and 10.71%/18.23%, respectively; the rest of the observed differentiation (33.69%/25.58%) was observed among populations.

No significant isolation by distance was detected by the Mantel test ($r = 0.0694$; $P = 0.7985$), suggesting that genetic differentiation in populations might not be the result of isolation by geographic distance (Fig. 3).
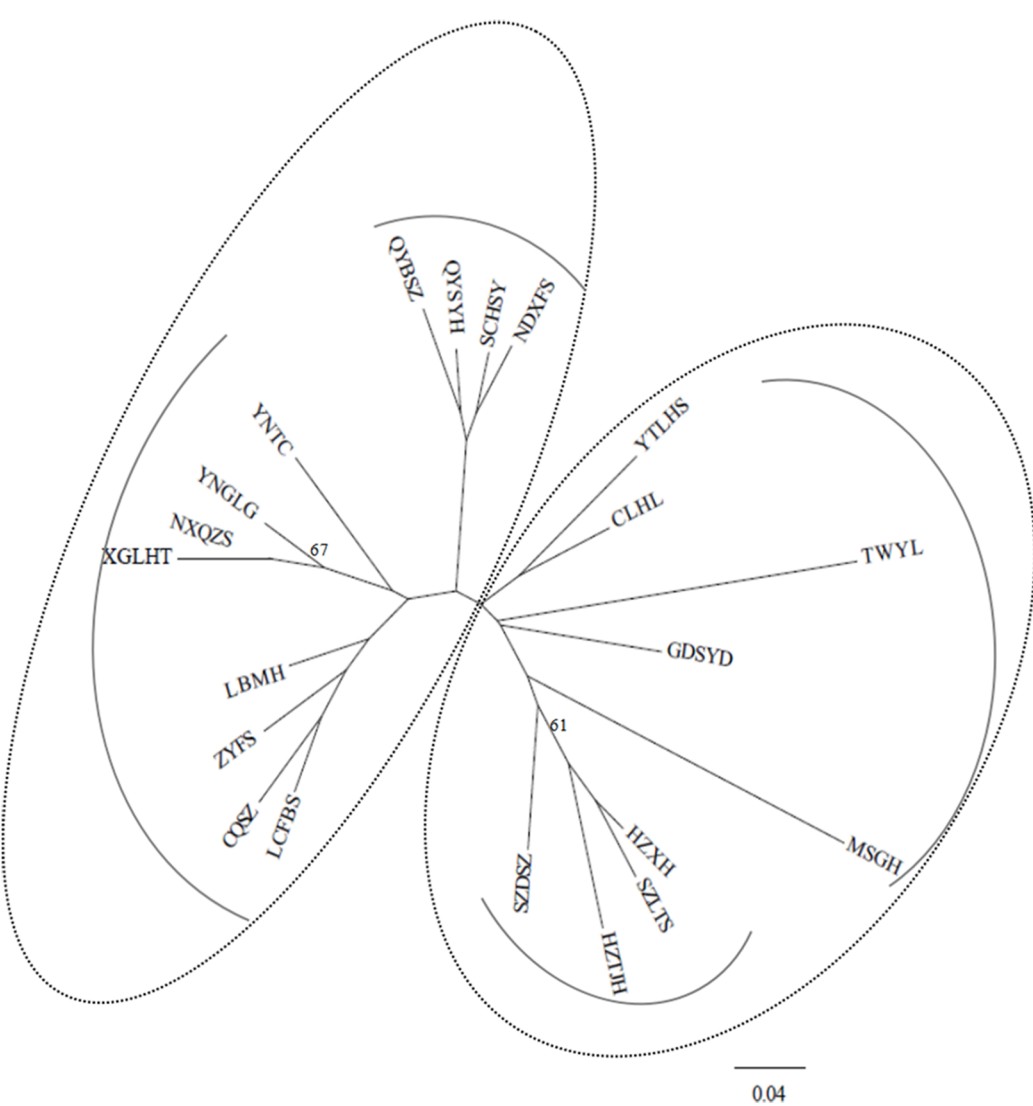

0.04

**Figure 2 NJ tree based on genetic distances (*Nei, 1987*) of 21 studied populations of *B. schreberi.***

**Table 4 AMOVA in *B. schreberi* based on 12 SSR loci.**

| Source of variation | DF | SS | VC | PV (%) | *P* value | |
|---|---|---|---|---|---|---|
| Among populations | 20 | 824.767 | 1.109 | 41.32 | <0.001 | $F_{ST} = 0.413$ |
| Within populations | 731 | 1,150.575 | 1.574 | 58.68 | <0.001 | |
| Among groups | 1 | 146.867 | 0.303 | 10.71 | <0.001 | $F_{CT} = 0.107$ |
| Among populations within groups | 19 | 677.900 | 0.954 | 33.69 | <0.001 | $F_{SC} = 0.377$ |
| Within populations | 731 | 1,150.575 | 1.574 | 55.60 | <0.001 | $F_{ST} = 0.444$ |
| Among groups | 3 | 364.146 | 0.511 | 18.23 | <0.001 | $F_{CT} = 0.182$ |
| Among populations within groups | 17 | 460.620 | 0.716 | 25.58 | <0.001 | $F_{SC} = 0.313$ |
| Within populations | 731 | 1,150.575 | 1.574 | 56.19 | <0.001 | $F_{ST} = 0.438$ |

**Note:**
DF, degrees of freedom; SS, sum of squares; VC, variance components; PV, percentage of variation.

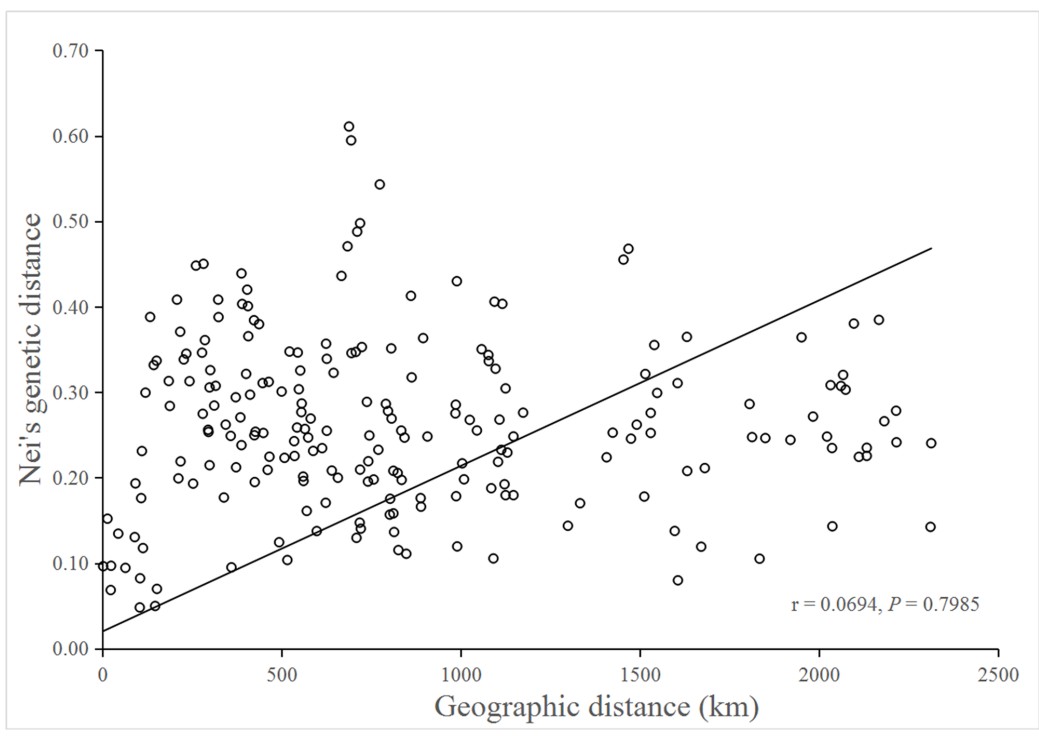

**Figure 3 Mantel test for matrix correlation between Nei's genetic distance and geographic distance of 21 *B. schreberi* populations.**

## DISCUSSION

Analyses of polymorphisms of 12 SSR loci in 376 individuals from 21 populations of *B. schreberi* in China revealed low levels of genetic diversity within populations (mean $H_E = 0.256$, range 0.139–0.398) compared with other aquatic plants but a remarkable genetic structure among populations. For example, using 10 SSR markers, *Chen et al. (2012)* revealed high levels of genetic diversity (mean $H_E = 0.610$, range 0.572–0.636) across seven studied populations of *Zizania latifolia* from central China; *Zhai, Yin & Yang (2018)* using eight SSR markers detected a moderate level of genetic diversity (mean $H_E = 0.351$, range 0.125–0.578) in 183 individuals of *O. acuminata* sampled throughout the species distribution. Two genetic clusters were revealed in *B. schreberi* in China by both Bayesian clustering and NJ cluster analyses, but with low bootstrap supporting values in NJ dendrogram, which were probably as a result of low genetic variation existed among populations. Within each genetic cluster, two subclusters were also detected, and the major genetic subclusters generally coincided with geographical regions. Several factors such as long-term survival in glacial refugia with some population expansion or long-distance dispersal, introgression, and population fragmentation may have contributed to the population genetics structure of *B. schreberi* in China.

Analysis of molecular variation indicated substantial among-population differentiation ($F_{ST} = 0.413$, $P < 0.001$) of *B. schreberi* in China. Furthermore, Mantel testing did not detect any significant isolation by distance ($r = 0.0694$, $P = 0.7985$), thus suggesting that

geographic distance might not be the main factor causing genetic differentiation among *B. schreberi* populations. The significant genetic differentiation found in *B. schreberi* populations can probably be explained by its long evolutionary history in glacial refugia. The current populations of *B. schreberi* in China are mostly located in biodiversity hotspots and centers of endemism for plants in China; for example, populations YNGLG and YNTC, LBMH and MSGH, GDSYD, CLHL, YTLHS, XGLHT, and NXQZS are distributed in the eastern Himalayan region, the Hengduan Mountain region, and the Nanling region, respectively, three of the world's biodiversity hotspots (*Myers et al., 2000*). Populations CQSZ and LCFBS were located in the "Eastern Sichuan-Western Hubei" region, an endemic area for Chinese flora. These regions were not directly affected by repeated Pleistocene glaciations; thus, these regions may have served as major glacial refugia for plant species in China (*Wang & Ge, 2006*; *Qiu, Fu & Comes, 2011*). The great diversity in topography, climate, and ecological conditions of these regions may have provided opportunities for population differentiations of *B. schreberi*. This pattern of extensive population differentiation has also been revealed in various plant species that are distributed in similar geographic regions as *B. schreberi* (*Qiu et al., 2009*; *Lei et al., 2012*; *Shi et al., 2014*; *Sun et al., 2014*). *Lei et al. (2012)* investigated the phylogeography of the Chinese beech, *Fagus engleriana*, in subtropical China using sequences of two chloroplast intergenic spacers, which indicated that most genetic variations occur among populations ($G_{ST} = 0.831$, $N_{ST} = 0.855$), suggesting that long-term isolation of *F. engleriana* populations among multiple refugia during the Pleistocene climatic changes might be the main driving factor for its population divergence. *Shi et al. (2014)* studied the phylogeography of *Castanopsis eyrei*, a widely distributed tree of subtropical evergreen broad-leaved forests in China. In their study, both nuclear microsatellite and cpDNA data indicated high levels of population differentiation (SSR: $F_{ST} = 0.443$; cpDNA: $G_{ST} = 0.709$, and $N_{ST} = 0.729$). They observed that the patterns of genetic differentiation between the extant populations of *Castanopsis eyrei* may have been affected by topographical differences between the mountainous western and lowland eastern refugia.

STRUCTURE and NJ tree analyses indicated that *B. schreberi* populations were comprised of two distinct genetic clusters. One of the clusters included a total of 12 populations, which were further divided into two subclusters. Another cluster included the other sampled populations, which were also separated into two subclusters. These two major genetic clusters were not always grouped by geographical region; for example, the populations of NXQZS, XGLHT, QYBSZ, QYSYH, SCHSY, and NDXFS of cluster II are located in Southeast China (Nanling regions), whereas the rest of the populations in this cluster are located in Southwest and Central China. Specifically, populations NXQZS and XGLHT from the Southeast China region were subclustered with the populations from the Southwest and Central China regions (e.g., populations YNTC, YNGLG, LBMH, CQSZ, LCFBS, and ZYFS). Population TWYL from Taiwan Island was included in cluster I despite being geographically separated from the other populations in the Nanling region (e.g., the populations of MSGH, GDSYD, CLHL, and YTLHS). However, most of the subclusters coincided with specific geographical regions. For example,

populations QYBSZ, QYSYH, SCHSY, and NDXFS from Zhejiang and Fujian provinces, with their short geographic distances, formed a subcluster. Similar geographical groups were found in populations HZTJH, HZXH, SZLTS, and SZDSZ from Zhejiang and Jiangsu provinces, and populations MSGH, GDSYD, CLHL, and YTLHS from Hunan, Guangdong, and Jiangxi provinces. The admixture pattern of populations of the two main genetic clusters of *B. schreberi* revealed in this study may be the result of long-distance dispersals. However, considering the highly fragmented modern populations of *B. schreberi* in China, we would not propose contemporary long-distance migrations of this species; instead, we suggest that the long-distance dispersal events are likely to represent historical migration events such as the Pleistocene interglacial/postglacial expansions from different refugia.

During the Pleistocene, the climate in subtropical China was characterized by significant cold–warm alternations, which have significantly affected plant population dynamics (*Qi et al., 2012*). Uplifting of mountains in the subtropical regions of China began in the Pleistocene (*Ming, 1987*; *Tang et al., 1994*; *Luo et al., 2010*). For example, the uplifting of the Yunnan–Guizhou Plateau began in the early Pleistocene and had a fast and dramatic uplift in the middle Pleistocene (*Tang et al., 1994*); furthermore, the Wuyi mountains were uplifted in the late Pleistocene (*Luo et al., 2010*). During periods of weak tectonic movements and/or more favorable climate (e.g., the Pleistocene interglacial periods), there is evidence indicating that numerous plant species underwent range expansions in subtropical China such as *Sagittaria trifolia* (*Chen et al., 2008*), *Cercidiphyllum japonicum* (*Qi et al., 2012*), *Pteroceltis tatarinowii* (*Li et al., 2012*), *Tetracentron sinense* (*Sun et al., 2014*), *Quercus glauca* (*Xu et al., 2015*), and *Sargentodoxa cuneata* (*Tian et al., 2015*). As a clonal aquatic herb with both sexual and asexual reproductive modes, *B. schreberi* should have been abundant enough to disperse in subtropical China during these periods. The distinct TWYL population on the island of Taiwan may have emerged after long-distance dispersal from the adjacent continent because land bridges between Taiwan and the Asian mainland formed during the Pleistocene, in line with changes in sea levels (*Huang & Lin, 2011*). However, the subtropical regions of China were subsequently fragmented by the intense topographical changes involving mountains, which in turn may have brought about genetic drift and/or extinction of isolated populations.

Range expansions may result in low levels of genetic diversity in newly established populations through a founder effect or a bottleneck effect (*Excoffier, Foll & Petit, 2009*). Our investigation of genetic diversity and bottlenecks in *B. schreberi* populations in China revealed low levels of genetic diversity within populations, and half of the studied populations might have experienced significant bottlenecks; this suggests that some of the *B. schreberi* populations in China might have been established by range expansion events. However, multiple range expansions from different refugee populations could trigger introgression or admixture and generate hybrid zones among populations in contact (*Excoffier, Foll & Petit, 2009*). Populations undergoing introgression and hybridization are mainly characterized by more admixed genotypes. STRUCTURE analysis of the *B. schreberi* populations showed that the genetic patterns of hybridization

and introgression potentially involve subcluster I-2 (e.g., populations SZDSZ, HZTJH, SZLTS, and HZXH) and subcluster II-2 (e.g., populations SCHSY, QYBSZ, QYSYH, and NDXFS). A high number of individuals from these two subclusters and from populations TWYL and CLHL contained admixed genotypes, with their $Q$ values supporting membership in both clusters. The populations from the subcluster I-2 and subcluster II-1 locations and the populations TWYL and CLHL might be the result of northward or eastward expansions of the southern, western, or central China populations. Similar dispersals such as the northward expansion from the Nanling Mountains along the Luoxiao Mountains to Dabie Mountain and the north-eastward expansion from the Nanling Mountains to the mountains of eastern Zhejiang and Fujian provinces along the coastline are evidenced by the extensive distribution of woody deciduous tree, *Sargentodoxa cuneata* (*Tian et al., 2015*).

## CONCLUSION AND IMPLICATIONS FOR CONSERVATION

Preservation of genetic variation is a principal target of conservation biologists for endangered species because it is critically important to maintain their evolutionary potential to survive in an ever-changing environment (*Shao et al., 2015*; *Hu et al., 2010*). To our knowledge, this is the first study that includes almost all known natural populations of *B. schreberi* in China, and therefore, the results generated in this study may facilitate the development of an effective conservation program for this species in China and other areas around the world. Compared to other endemic aquatic plants, this species shows low genetic diversity and significant genetic differentiation. The long-term evolutionary history in glacial refugia, the introgression following the range expansions, and the habitat fragmentation might have played a vital part in the genetic differentiation of this species. In China, *B. schreberi* comprises four genetically distinct subgroups, which should be treated as separate units in conservation programs. For an ex situ conservation procedure, sampling more regions where the distinct genetic subgroups are located is suggested to cover more genetic diversity. Recent anthropogenic intervention through polluted water environments or deforestation might be the principal factor that is responsible for the current endangered state of *B. schreberi*. Its habitat has been damaged by human activities, including the widespread use of herbicides in agriculture (*Dong, 2010*) and overexploitation for profit. Essentially, in our field investigation and in previous reports (*Dong, 2010*), water pollution has worsened in habitats near residential areas. In addition, intense habitat fragmentation has also occurred in the field. Hence, protection of the remaining habitats of *B. schreberi* should be a priority of conservation programs.

Although great efforts have been made to reveal the population genetic structure of *B. schreberi* in China, samples from other regions around the world are limited. In addition, our DNA sequence-based analyses of the phylogeographic structure of *B. schreberi* revealed low polymorphism in the nuclear internal transcribed spacer (ITS) region and 20 chloroplast non-coding regions in six individuals sampled from different populations in China (Fig. S1). Thus, the evolutionary history of *B. schreberi* in China remains unclear. Additional samples from other places around the world and high

throughput next-generation sequencing techniques (e.g., restriction-site-associated DNA sequencing or genotyping-by-sequencing) can be used to construct a more extensive phylogeographic history of *B. schreberi* in China, which could lead to more robust conservation strategies.

## ACKNOWLEDGEMENTS

The authors thank Hui Xu, Meng-Xue Lu, Ying Zhang and Shi-Xu Huang for their assistance in the laboratory, and Wen Huang, Jing Xia, and Hong Liu for their help in fieldwork.

### Funding

This work was supported by grants from the Strategic Priority Research Program of the Chinese Academy of Sciences (No. XDB31000000), Special Foundation for State Basic Working Program of China (No. 2013FY112300), the National Natural Science Foundation of China (No. 31570220) and Wuhan Botanical Garden (CAS) (No. Y655261W03). The funders had no role in study design, data collection and analysis, decision to publish, or preparation of the manuscript.

### Grant Disclosures

The following grant information was disclosed by the authors:
Strategic Priority Research Program of the Chinese Academy of Sciences: XDB31000000.
Special Foundation for State Basic Working Program of China: 2013FY112300.
National Natural Science Foundation of China: 31570220.
Wuhan Botanical Garden (CAS): Y655261W03.

### Competing Interests

The authors declare that they have no competing interests.

### Author Contributions

- Zhi-Zhong Li conceived and designed the experiments, performed the experiments, analyzed the data, contributed reagents/materials/analysis tools, prepared figures and/or tables, authored or reviewed drafts of the paper, approved the final draft.
- Andrew W. Gichira analyzed the data, contributed reagents/materials/analysis tools, prepared figures and/or tables, authored or reviewed drafts of the paper, approved the final draft.
- Qing-Feng Wang contributed reagents/materials/analysis tools, approved the final draft.
- Jin-Ming Chen conceived and designed the experiments, analyzed the data, authored or reviewed drafts of the paper, approved the final draft.

### Data Availability

  The raw data are provided in the Supplemental Files.

## Supplemental Information

Supplemental information for this article can be found online at http://dx.doi.org/10.7717/peerj.5296#supplemental-information.

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
