# Peer review of "Genetic diversity and population structure of the endangered basal angiosperm Brasenia schreberi (Cabombaceae) in China"

_PeerJ, doi:10.7717/peerj.5296_

## Round 0.1 · original submission · Minor Revisions

Although I have suggested Minor Revisions, these revisions are essential to the paper. You should pay particular attention to the suggestions for revision of the reported statistics. One reviewer has suggested the addition of plastid markers and I would support this as a means of making the paper much stronger, however, it is not essential as the nuclear markers do offer a good initial insight into the population variation. I do plan to send the paper back to these reviewers once the modifications have been made.

Reviewer 1 ·

Basic reporting

1. Clear and unambiguous, professional English used throughout. – ok
2. Literature references, sufficient field background/context provided – I missed more references on genetic diversity and structure in macrophytes. Since Brasenia schreberi is an aquatic plant, I think this is crucial
3. Professional article structure, figs, tables. Raw data shared – ok
4. Self-contained with relevant results to hypotheses – I don’t reckon the work is self-contained yet. I think the results of other molecular markers in the manuscript are important.

Experimental design

1. Original primary research within Aims and Scope of the journal. – ok
2. Research question well defined, relevant & meaningful. It is stated how research fills an identified knowledge gap. – ok
3. Rigorous investigation performed to a high technical & ethical standard. – I reckon that the discussions and conclusions that the authors provide will become more sound if the work had used some other molecular marker, for instance, a plastidial one. Nuclear Microsatellites are excellent tools to solve more recent issues in historical time and along with such markers may contribute greatly to the understanding of older evolutionary aspects. To draw sound conclusions about the phylogeography of plants during glaciations periods a plastid marker is crucial. If we analyze the studies quoted by the authors in the Discussion most of them used at least one plastidial marker, none used only SSR. Moreover, the vast majority of these works were done by having trees or shrubs as focus of the study. Almost no comparison was made with other aquatic plants by the authors of the manuscript. This is also critical to the work's soundness. I should mention also that according to prevailing ethical standards, information on submission of DNA sequences to an appropriate repository (such as GenBank) must be given. Voucher specimens information must also be clearly provided in text (herbarium code). And these informations were not clearly quoted in the manuscript.
4. Methods described with sufficient detail & information to replicate. – As regards SSRs markers, since incomplete information were provided by the authors, it is not possible to reproduce the experiments.

Validity of the findings

1. Impact and novelty not assessed. Negative/inconclusive results accepted. Meaningful replication encouraged where rationale & benefit to literature is clearly stated. – ok
2. Data is robust, statistically sound, & controlled. – the data were not sound enough to support the conclusions.
3. Conclusion are well stated, linked to original research question & limited to supporting results. – The conclusions are not limited to those supported by the results.
4. Speculation is welcome, but should be identified as such. - ok

Additional comments

I believe the authors have a potentially excellent work at hand. Studies that assess genetic diversity in aquatic plants are lacking. And the conservation of these plants is a primary issue. Microsatellites are excellent markers that provide a wealth of information capable of answering many questions about plant biology. However, I believe that it would be important to adapt the discussion and conclusions to the questions that microsatellites can resolve with soundness. In order to assess the phylogeographical issues that authors wish to discuss, using another marker would be important. I strongly recommend that the authors make the necessary adaptations and resubmit the work.

Reviewer 2 ·

Basic reporting

The authors present a paper on the population genetics of Brasenia schreberi. The article is very well written and I have no hesitation in recommending it to be published, providing that typos and inconsistencies in spelling and formatting are corrected.

Experimental design

See below

Validity of the findings

The experimental design and analysis of the data set is thorough and I found the results interesting.

Additional comments

Line 39 I think this should be ‘Microsatellites’
Line 51 leaves during food gathering is confusing. Do you mean leaves harvested for food?
Line 65 decrease in the size of
Line 96 The CTAB reference should be Doyle & Doyle (1987). It is correct in the References.
Line 116-128 The styling of genetic diversity measures is inconsistent, e.g. line 116 Na vs. line 121 NA
Line 121 Did you mean ‘number of private alleles (Ap)’?
Line 121 If the explanation of NA is included the other measures should be as well, or only the ones that are not listed in the earlier paragraph
Line 139 Did you mean ‘with 70%’?
Line 145 I think it would be clearer as: (K) was tested from K=2 to 20 clusters, based on…
Line 148 structure harvester is confusing as this is a name of a program. It might be clearer to delete structure harvester, i.e. was verified based on the
Line 187-188 You mention one subgroup, what about the other subgroups?
Line 210 The P value and the test of significance in AMOVA relates to the probability of the Fst value being different from random, but does not test biological significance. I think it would be best to replace significant with substantial in the sentence.
Line 288 ‘among populations in contact’ would be better
Line 300 evidenced instead of evidence
Line 309 Delete ’in’ i.e. facilitate the development
Line 316 Should be: subgroups are located
Line 328 Did you mean 20 cpDNA regions in total? In which case total is not necessary.
Line 333-335 The last sentence is near verbatim repeat of line 309-310. I think extra data both in terms of sampling and NGS methods could lead to more robust conservation strategies.
Table 2 Headings are inconsistently formatted, some in subscript some not
Table 3 I have not found *** significance in the table. Either the asterisk is missing in the table or there is no need to list it in the caption.
Table 4 whitin is misspelled. Among groups appears the second time without capitals. It would be useful to indicate how this table is actually for the results of three separate AMOVA’s. Maybe some form of subdivision of the table could help?
Figure 1 Some labels on the map do not match the text of the paper e.g. COSZ/CQSZ, XGLH/XGLHT
Figure 3 I don’t think there is any point starting the X axis at -1000 km. Maybe start is just below 0 so no dots fall on the Y axis. The X axis label is missing a space between distance and (km)

·

Basic reporting

The manuscript describes the genetic diversity and population structure of the endangered basal angiosperm Brasenia schreberi (Cabombaceae) This a very important conservation effort and good scientific merit.

Experimental design

The aim of the study is fulfilled.
The intent and the composition of the paper are good:
- The sampling strategies are correct to reach the desired goals (21 populations and 376 individuals).
- 12 microsatellite markers used provide good amount of information.
- Statistical analysis is satisfied.
- Figures and table are representative.
- The conclusion is also meaningful.

Validity of the findings

The paper is nearly ready to publish. Furthermore there are some points for consideration.

Additional comments

The authors could consider the following suggestions:

1- In the summary statistics for B. schreberi populations in China (Table 3) you must provide confidence interval for Ar, Ho, He and Fis. Standard deviations is not the best statistical approach to compare populations using its parameters.
2- Provide more information about null alleles.
3- What is the effect of the low bootstrap nodes in NJ dendrogram?
4- BOTTLENECK is flexible and results depends on the chosen parameters. Cite and provide more information about it.

---

## Round 0.2 · Minor Revisions

You have made substantial improvements to the manuscript and the reviewers have identified a number of further revisions and corrections needed. Please make these and then your article will be acceptable for publication.

Reviewer 1 ·

Basic reporting

1. Clear and unambiguous, professional English used throughout. – ok
2. Literature references, sufficient field background/context provided – revised!
3. Professional article structure, figs, tables. Raw data shared – ok
4. Self-contained with relevant results to hypotheses – explained!

Experimental design

1. Original primary research within Aims and Scope of the journal. – ok
2. Research question well defined, relevant & meaningful. It is stated how research fills an identified knowledge gap. – ok
3. Rigorous investigation performed to a high technical & ethical standard. – explained! It seems ok for me now.
4. Methods described with sufficient detail & information to replicate. – revised!

Validity of the findings

1. Impact and novelty not assessed. Negative/inconclusive results accepted. Meaningful replication encouraged where rationale & benefit to literature is clearly stated. – ok
2. Data is robust, statistically sound, & controlled. – explained! It seems ok for me now.
3. Conclusion are well stated, linked to original research question & limited to supporting results. – explained!
4. Speculation is welcome, but should be identified as such. – ok

Additional comments

In this version of the work the authors clarified many important things, which I missed in the previous version. I believe it can be recommended to publish now. I just want to highlight a few simple points for consideration below:
Line 41 - Bertin, 1993 is¬ missing in References.
Line 76 – Martínez instead of Martinez.
Line 77- Nuphar submersa instead of Nuphar submerse.
Line 116 - Yeh et al., 1999 must be in italics.
Line 139 - check the punctuation after “IAM”.
Line 153 - why “e.g.”? Is there any other locus in which null alleles were observed?
Line 156 – the 5.08 value doesn’t match with the value observed in Table 02.
Line 158 – the 0.414 value doesn’t match with the value observed in Table 02.
Line 159 – the value 0.223 doesn’t match with the value observed in Table 02.
Line 160 – Table 02 instead of Table S1?
Line 166 – It is mentioned “HO…0.310 ± 0.120”. But, what about TWYL population? HO = 0.354(0.137).
Line 176 – Isn’t it nine populations instead of eight?
Line 179 - Isn’t it two subclusters instead of four?
Line 180 – I think it is cluster II instead of cluster 2.
Line 182-183 – I think the sentence “It is possible that the clusters were correlated with geographic location and different levels of introgression among various populations” fits better in Discussion.
Line 222 - Wang & Ge 2006 is missing in References.
Line 226 – “Qiu et al., 2009”. Is this Qiu et al. 2011? Or a work that is missing the References?
Line 243 – Instead of Cluster I is Cluster II, at least according to Fig_1.
Line 263-264 - Ming, 1987 is missing in References.
Line 269-270 - Chen et al., 2008 is missing in References.
Line 270 - Li et al., 2012 is missing in References.
Line 290 - Subcluster I-2 according to Fig_1.
Line 291 - Subcluster II-2 according to Fig_1.
Line 304 – “Shao et al., 2009”. Is it Shao et al. 2015 (mentioned in References) or another work?
Line 304 - Hu et al., 2010 is missing in References.
Line 307-309 – “Compared to other endemic aquatic plants, this species shows low genetic diversity and significant genetic differentiation.” It would be nice if this comparison were better discussed in Discussion section.
Line 376 – This reference is not mentioned in the text.
Line 459-461 – This reference is not mentioned in the text.
Line 498- The populations SZDSZ and YNGLG are incorrectly typed in the Figure.
In the References Section some of the references are out of order and there are some typos.

Reviewer 2 ·

Basic reporting

OK

Experimental design

OK

Validity of the findings

OK

Additional comments

I am happy with the corrections the authors made to the paper and recommend it to be published. I would change ‘significant’ to ‘substantial’ in line 209 to be consistent with line 191.

·

Basic reporting

Many of the requested changes have been incorporated, but there are several key points in the reviewed manuscript:
What is the effect of the low bootstrap nodes in NJ dendrogram? Use your response in the discussion of NJ dendrogram.

Please cite the BOTTLENECK chosen parameters.

Line 205: “It is possible that the clusters were correlated with geographic location and different levels of introgression among various populations.”
Line 223: “that genetic differentiation in populations might not be the result of isolation by geographic distance”
It is contradictory, why?

Experimental design

The aim of the study is fulfilled.
The intent and the composition of the paper are good:
- The sampling strategies are correct to reach the desired goals (21 populations and 376 individuals).
- 12 microsatellite markers used provide good amount of information.
- Statistical analysis is satisfied.
- Figures and table are representative.
- The conclusion is also meaningful.

Validity of the findings

The paper is nearly ready to publish. Furthermore there are some points for consideration.

Additional comments

Many of the requested changes have been incorporated, but there are several key points in the reviewed manuscript:

What is the effect of the low bootstrap nodes in NJ dendrogram? Use your response in the discussion of NJ dendrogram.


Please cite the BOTTLENECK chosen parameters.


Line 205: “It is possible that the clusters were correlated with geographic location and different levels of introgression among various populations.”
Line 223: “that genetic differentiation in populations might not be the result of isolation by geographic distance”
It is contradictory, why?

In the summary statistics for B. schreberi populations in China (Table 3) you must provide confidence interval for Ar and Fis. Use diveRsity R package.

---

## Round 0.3 · accepted · Accept

Thank you for your comprehensive response to the reviewers feedback. I am satisfied you have addressed all the outstanding issues.

#